# Vascularisation in Deep Endometriosis: A Systematic Review with Narrative Outcomes

**DOI:** 10.3390/cells12091318

**Published:** 2023-05-05

**Authors:** Simon G. Powell, Priyanka Sharma, Samuel Masterson, James Wyatt, Ilyas Arshad, Shakil Ahmed, Gendie Lash, Michael Cross, Dharani K. Hapangama

**Affiliations:** 1Department of Women’s and Children’s Health, Institute of Life Course and Medical Science, University of Liverpool, Liverpool L8 7SS, UK; simon.powell@liverpool.ac.uk (S.G.P.);; 2Liverpool Women’s Hospital NHS Foundation Trust, Liverpool L8 7SS, UK; 3Liverpool University Hospitals NHS Foundation Trust, Liverpool L7 8XP, UK; 4School of Medicine, University of Liverpool, Liverpool L8 7SS, UK; 5Guangzhou Institute of Pediatrics, Guangzhou Women and Children’s Medical Center, Guangzhou Medical University, Guangzhou 510180, China; 6Department of Pharmacology and Therapeutics, Institute of Systems, Molecular and Integrative Biology, University of Liverpool, Liverpool L69 3GE, UK

**Keywords:** deep endometriosis, endometriosis, systematic review, vascularisation, microvessel density, VEGF-A, HIF-1A

## Abstract

Deep endometriosis (DE) is the most severe subtype of endometriosis, with the hallmark of lesions infiltrating adjacent tissue. Abnormal vascularisation has been implicated in contributing to endometriosis lesion development in general, and how vascularisation influences the pathogenesis of DE, in particular, is of interest. This systematic review followed the PRISMA guidelines to elucidate and examine the evidence for DE-specific vascularisation. A literature search was performed using MEDLINE, Embase, PubMed, Scopus, Cochrane CENTRAL Library and Europe PubMed Central databases. The databases were searched from inception to the 13 March 2023. A total of 15 studies with 1125 patients were included in the review. The DE lesions were highly vascularised, with a higher microvessel density (MVD) than other types of endometriotic lesions, eutopic endometrium from women with endometriosis and control tissue. Vascular endothelial growth factor, its major subtype (VEGF-A) and associated receptor (VEGFR-2) were significantly increased in the DE lesions compared to superficial endometriosis, eutopic endometrium and control tissue. Progestin therapy was associated with a significant decrease in the MVD of the DE lesions, explaining their therapeutic effect. This review comprehensively summarises the available literature, reporting abnormal vascularisation to be intimately related to the pathogenesis of DE and presents potentially preferential therapeutic targets for the medical management of DE.

## 1. Introduction

Endometriosis is a chronic, non-cancerous gynaecological condition defined as the presence of endometrium-like tissue residing outside the uterine cavity [1]. It is the second-most prevalent gynaecological condition in the UK, affecting one in ten women [2]. The exact pathological mechanisms relevant to the growth and establishment of endometriotic lesions in ectopic sites remain unknown. Although a new unifying pathogenesis theory has been proposed, it is yet to gain acceptance with robust evidence [3].

Sampson’s theory of retrograde menstruation is the most accepted concept explaining the aetiology of endometriosis [4,5]. Sampson proposes that endometriosis occurs due to the retrograde flow of sloughed endometrial cells via the fallopian tubes into the pelvic cavity during menstruation [5,6,7,8]. Over 90% of women reflux endometrial debris into the peritoneal cavity during menstruation, yet endometriosis only develops in 10–15% of women [4,9]; it is unclear why some women develop endometriosis whilst others do not. It is postulated that several other mechanisms, such as immune dysfunction, including inflammation and the escape of ectopic endometrial tissue from immunological control, genetic and epigenetic factors and abnormal vascularisation may contribute to the development of endometriosis [4,10]. A wide variety of symptoms is associated with endometriosis, including chronic pelvic pain, dysmenorrhoea, dyspareunia, fatigue, infertility, depression and anxiety [11]. This extensive symptom profile devastates women’s quality of life and limits their economic productivity [12,13].

Endometriotic lesions are categorised into superficial peritoneal, ovarian endometrioma and deep endometriosis (DE). There is an ongoing debate regarding whether these distinct forms of endometriosis represent a single spectrum of disease or diverse pathways of pathogenesis [4].

DE is a severe subset of endometriosis in which endometrial tissue infiltrates more than 5 mm into the peritoneal surface or invades directly into intraperitoneal organs such as the bladder or bowel [14]. DE lesions are strongly associated with pelvic pain [15], and the severity of pain correlates with the infiltration depth and increased vascularisation [16].

Vascularisation is the process of new blood vessel formation and incorporates three distinct principle processes, which are (1) angiogenesis, the construction and remodelling of new blood vessels from pre-existing vessels [17,18]; (2) postnatal-vasculogenesis, the de novo generation of microvessels by bone marrow-derived endothelial progenitor cells (EPCs) [19]; and (3) inosculation, the incorporation of a implants’ pre-formed microvascular network into the host’s own microvascular network [20]. Since the exact mechanisms by which endometriotic lesions are formed are not fully understood, deciphering which of the vascularisation processes is involved in or specific to endometriotic lesions is challenging. At present, there are no curative treatments for endometriosis. We expect to facilitate the development of novel and effective therapies by generating a greater understanding of the underlying disease process of DE.

Vascularisation is central to the complex changes that occur in the accurately located (eutopic) endometrium during the menstrual cycle [21]. Abnormal vascularisation is also postulated to be critical in the development of endometriosis by establishing a blood supply to newly formed ectopically situated endometriotic lesions sanctioning the lesion to proliferate [22]. Previous studies have demonstrated excessive endometrial angiogenesis and abnormal vascularisation in DE lesions [22]. Two angiogenic factors which may play a role in the vascularisation of endometriotic lesions are the vascular endothelial growth factor (VEGF) and Hypoxia-Inducible Factor 1 Alpha (HIF-1A) [23].

VEGF is a family of glycoproteins with potent angiogenic properties implicated in both physiological and pathological angiogenesis [24,25]. VEGF comprises multiple isoforms, including VEGF-A, VEGF-B, VEGF-C, VEGF-D, VEGF-E and placental growth factor (PlGF). VEGF-A and its splice variant, VEGF-A165, are the most comprehensively studied isoforms within the VEGF family. VEGF-A is released by many cells in response to tissue hypoxia. The physiological, angiogenic response of VEGF-A is mediated primarily through the tyrosine kinase receptor VEGF-R2 expressed on vascular endothelium [26,27,28]. VEGF-A has three main functions in vascularisation, including promoting endothelial cell proliferation and migration, increasing microvascular permeability and the secretion of Matrix metalloproteinases (MMPs) [25,29].

HIF-1A protein is a transcription factor that regulates the transcription of multiple genes. HIF-1A is a master regulator of tissue response to intracellular hypoxia and is critical to the pathways involved in angiogenesis, haemopoiesis, cell proliferation, apoptosis and vasodilation [30,31,32]. HIF-1A causes increased neovascularisation in response to hypoxia through many downstream genes, including VEGF [33].

This systematic review aims to consolidate the published data on the association between vascularisation and DE. By doing so, we also aim to examine the underlying mechanisms of vascularisation in the pathogenesis of DE.

## 2. Materials and Methods

This systematic review follows the Preferred Reporting Items for Systematic Review and Meta-Analyses (PRISMA) guidelines [34]. A prospective protocol was registered with the International Prospective Register of Systematic Reviews (PROSPERO) database on 7 January 2022 (registration number CRD42022293688).

### 2.1. Systematic Search

A systematic search was performed using MEDLINE, Embase, PubMed, Scopus, Cochrane Library and Europe PubMed Central databases. All databases were searched from the database’s inception to the 13 March 2023.

The search string employed a combination of medical subject headings (MeSH) and free-text search terms for the two study components, “Deep Infiltrating Endometriosis” and “Mechanism of Vascularisation”. The search strategy is presented in Appendix A. Grey literature was not included in the search. All relevant papers, including non-English language results, were included and reviewed.

### 2.2. Eligibility Criteria

All human and animal studies reporting original data concerning the mechanism of vascularisation of endometriotic lesions in the pathophysiology of DE were included in this review. Relevant studies of all languages were included. Non-English language studies were translated into English using Google Translate, a neural machine-based translation service with established accuracy [35]. Studies were excluded if they were not published in an established journal with a peer-review process. Studies with non-adult participants were excluded, as were studies involving patients with a pelvic malignancy.

### 2.3. Study Selection

Results from the initial searches were collated, and duplicates were deleted. Title and abstract screening were completed independently by two authors (PS and SM) using the online manuscript screening tool Rayyan [36].

Full texts were retrieved and reviewed independently by PS and SM. Each study was evaluated for inclusion using the pre-determined eligibility criteria. Any disagreements were resolved via discussion among all authors, and a consensus was achieved. Additional studies were identified through forward and backward chaining of the included studies. The references of all the literature and systematic reviews identified by the initial search were also screened.

### 2.4. Data Extraction and Synthesis

Data were extracted independently by PS and SM. Data included, but were not limited to, title, author, journal, year of publication, population studied, interventions, phase of the menstrual cycle, hormonal therapy, results and outcomes.

The results were synthesised in a thematic manner. The authors independently identified recurring themes throughout the final list of included studies. This final list of themes was discussed and confirmed by the authors, encompassing the titles presented in the results section of this review. Meta-analysis was not possible in this review due to the heterogeneity of the methods and results across the included papers.

### 2.5. Bias Analysis

The Newcastle–Ottawa Scale was used to assess the quality of each study included in this review [37]. This scale gives a score out of nine based on three categories including selection, comparability and exposure. The categories are scored out of four, two and three, respectively. Scores of zero or one in selection or exposure highlight an increased risk of bias in these categories, while a score of zero in the comparability section gives a study a high risk of bias.

## 3. Results

The literature search identified 855 papers, of which 127 were duplicates and removed. A further 619 studies were excluded based on the title screening, and 51 were excluded based on the content of their abstracts. Of the 58 studies that underwent full-text review, 13 were deemed eligible and included in this review.

Citation screening was also undertaken on the 12 included studies, yielding 7 potentially relevant studies. Following a full-text review, two studies were found to be relevant and included. A total of 15 studies were included in this systematic review, all of which are original literature. The screening and chaining processes are summarised in Figure 1.

The selected 15 studies included 10,125 women of reproductive age. A variety of experimental methods was employed across the studies. The majority, eight, exclusively used immunohistochemical (IHC) staining to study the proteins of interest [38,39,40,41,42,43,44], and four exclusively used quantitative reverse transcription-polymerase chain reaction (qPCR) to assess the transcripts of the relevant genes [45,46,47,48,49]. Of the remaining two studies, one analysed peritoneal fluid using the ProSeek Multiplex Oncology Cancer Panel [50], and the other undertook an Enzyme-Linked Immunosorbent Assay (ELISA) analysis of the protein levels in peritoneal fluid and venous blood samples [51]. A detailed overview of this study’s characteristics is provided in Table 1 and Table 2.

The risk of bias was assessed using the Newcastle–Ottawa Scale (NOS). The NOS quality assessment scores for each individual study are detailed in Table 3.

### 3.1. Microvessel Density

Seven of the included papers investigated the microvessel density (MVD) in DE [38,39,40,41,42,43,46]. The endothelium of the blood vessels was stained using CD31, CD34, FSHR and vWF antibodies and an overall MVD was calculated by assessing the number of stained vessels per field in a given “endometriotic hot spot”.

Five studies compared the endometriotic lesions collected prospectively from patients undergoing laparoscopic endometriosis surgery to control tissue [38,39,41,46,53,54]. The control tissue was obtained either from the same patient at a disease-free site or tissue collected from patients undergoing laparoscopic surgery for benign diseases other than endometriosis. All studies reported an increased MVD in the DE lesions compared to eutopic endometrium.

Two studies compared DE and superficial endometriosis [41,46]. Jondet et al. reported DE lesions to have a higher MVD than other types of endometriotic lesions. In contrast, Robin et al. identified no significant difference between the endometriosis subgroups. The studies’ authors undertook contrasting staining techniques, with Jondet et al. using the endothelium marker CD31 and Robin et al. using FSHR, potentially accounting for the inconsistent findings.

Five studies compared paired ectopic and eutopic endometrial samples from patients with DE [39,40,41,43,46]. All five studies reported an increased MVD in ectopic endometrial lesions compared to eutopic endometrium.

Ectopic endometrium from patients with DE was directly compared to the control tissue obtained from healthy volunteers free from endometriosis in five studies, all of which reported an increased MVD in the DE lesions [38,39,41,43,46].

Two studies explored the impact of the menstrual cycle phase on the MVD in healthy eutopically located endometrium [41,43]. Both studies identified a higher MVD in the proliferative phase of the menstrual cycle compared to the secretory phase.

Jondet et al. investigated the impact of progestin treatment on the MVD in 66 patients with DE and superficial endometriosis. Using IHC and the endothelial cell marker CD31, the authors calculated the MVD, expressed as vessels per mm^2^, in patients with and without progestin treatment. Progestin treatment led to a statistically significant reduction in the MVD in both eutopic and ectopic lesions. Of note, progestin treatment had a lower effect in reducing the MVD in the DE lesions compared with superficial endometriotic lesions (19% reduction versus 31%, *p* = 0.0018) [46].

In a large study comprising 194 patients, Robin et al. compared the density of FSHR-positive blood vessels in the “core” of a DE lesion with the surrounding normal “host” tissue using IHC and the FSHR antibody. Researchers found that the MVD in the core of the DE tissue was two-fold higher than that of the adjacent normal host tissue (64.2 ± 8.2 vs. 27.2 ± 3.2 vessels/mm^2^). Researchers also identified a statistically significant positive correlation between the expression of FSHR in endometriosis glands and the duration of infertility (*p* = 0.0005) and the rectovaginal nodule size (*p* = 0.002) [41].

Raimondo et al. employed a two-stage approach to the evaluation of the MVD. Researchers in this study initially used intraoperative indocyanine green angiography (IGA) to allow macroscopic evaluation of the presence of different vascular patterns in 30 patients with symptomatic DE. IGA is an established technique commonly utilised in gynaecological surgery and has been found to be a valuable and safe technique for assessing the perfusion of intra-peritoneal organs in real-time. Perfusion grade was classified as follows: 0–1 = no or low fluorescence (hypovascular pattern); 2 = regular fluorescence, similar to healthy surrounding rectosigmoid mesentery (isovascular pattern); and 3–4 = diffuse or abundant fluorescence (hypervascular pattern). A total of 60% of the DE lesions were found to be hypovascular, while the remaining 40% were deemed to be hypervascular. Following the resection of the endometriotic lesion, the authors used IHC and CD31 staining to assess the MVD in five randomly chosen fields within the resected sample. The hypovascular lesions had a significantly lower mean MVD than the hypervascular lesions (154.6 vessels/mm^2^ vs. 281.1 vessels/mm^2^, *p* = 0.01). The hypovascular nodules had a larger maximum diameter (39.5 ± 15.5 mm, *p* < 0.05) and a lower MVD (154.6 ± 43.6 mm, *p* < 0.05) than the hypervascular nodules [42].

In a cohort of 113 patients with DE, Vinci et al. evaluated the MVD measured using IHC by assessing the number of CD31 stained vessels per field in a given “endometriotic hot spot”. CD31 staining was found to be “intense” in 42% of the DE tissue samples, “medium” in 32% and “weak” in 26% [38]. Stratopoulou et al. conducted an IHC investigation of rectovaginal DE (n = 13), uterine adenomyosis (n = 14) and control endometrial tissue collected from disease-free patients (n = 14). Vascular endothelium was demarcated with CD31 antibodies, and alpha-smooth muscle actin (αSMA) was used to stain smooth muscle cells in larger, more mature blood vessels. The authors demonstrated a significantly higher MVD in the DE lesions than in the healthy control tissue (*p* = 0.013). The ectopic DE tissue also exhibited a significantly higher MVD than its corresponding eutopic endometrium (*p* = 0.002). Although there was an increased number of vessels in the DE lesions compared to eutopic endometrium, only a small number of vessels stained positively for αSMA in the DE lesions. The rates of positive staining with αSMA were significantly lower in the DE lesions compared to eutopic endometrium from patients with DE (*p* = 0.001) and control endometrium (*p* = 0.002) [39].

Machado et al. used IHC to analyse the number of vWF-positive vessels in the DE lesions (n = 10) and control healthy endometrium from women without endometriosis (proliferative n = 10, secretory n = 10) and normal rectum (n = 4). The number of vWF-positive vessels was significantly increased in the DE lesions (16.5 ± 0.53/mm^2^) when compared with control proliferative (5.0 ± 0.74/mm^2^ (*p* < 0.05) and secretory (3.5 ± 0.70/mm^2^, *p* < 0.05) endometrium. vWF-positive vessel density was also significantly higher in the DE lesions (16.5 ± 0.53/mm^2^) compared to normal rectal tissue (10.4 ± 00.89/mm^2^, *p* < 0.05) [43].

Signorile et al. evaluated the hormone influence on vascularisation by analysing the estrogen (ER) and progesterone (PR) receptor levels in DE tissue and correlating them to the level of vascularisation. The authors assessed vascularisation using IHC and CD34 as a marker of vascular endothelial progenitors. Out of the 62 patient samples examined, 27 had an “intense” expression of CD34 and only 8 had “absent to very low” CD34 staining. ER and PR were expressed in all the tissue samples studied. The authors concluded that the ER and PR expression levels in the DE lesions correlated with the level of vascularisation (*p* = 0.001) [40].

Using IHC, Robin et al. evaluated the expression of the follicle-stimulating hormone receptor (FSHR) in 194 tissue specimens obtained from patients with superficial endometriosis, and ovarian endometrial cysts and DE Endometriotic lesions were compared to eutopic endometrium as a control tissue. The authors found a prominent FSHR-positive vascular pattern in all the endometriosis lesions. FSHR was not expressed in the surrounding healthy tissues located more than 5 mm from the endometriotic lesion. The density of the FSHR-positive vessels in the DE lesions was 46.0 ± 5.7 vessels/mm^2^, with no significant difference in density when compared to superficial or ovarian endometriosis [41].

### 3.2. Vascular Endothelial Growth Factor (VEGF) and Angiogenesis-Related Signalling Molecules

Ten studies investigated the VEGF glycoproteins and their associated receptors, with the majority of studies investigating VEGF-A [39,43,44,45,47,48,49,50,51]. Compared to superficial disease, VEGF was increased in four studies [43,45,50,52], reduced in two studies [47,49], and no difference was identified in two studies [39,44].

Four studies compared VEGF between ectopic and eutopic endometrium in women with DE. All three studies identified an increased expression of VEGF [43,45,49,52].

Compared to the control endometrial tissue, increased VEGF was reported in four studies [45,49,51,52], while one study identified no difference [47].

Ramon et al. analysed angiogenesis-related microRNAs (miRNAs), VEGF-A and thrombospondin-1 (TSP-1), using real-time polymerase chain reaction (RT-PCR). The authors further quantified the protein levels of VEGF-A and TSP-1 using ELISA. A total of 96 patients were prospectively recruited, including 58 women with endometriosis and 38 healthy women undergoing surgery as part of management for benign gynaecological conditions. The patients in the control group were proven to be free of endometriosis at the time of laparoscopy, and the samples were collected in the proliferative and secretory phases of the menstrual cycle. The authors compared paired eutopic and ectopic endometrium in women with DE and between ectopic endometrium and control eutopic endometrium obtained from patients free from endometriosis. The expression of several angiogenesis-related miRNAs was significantly increased in the DE lesions compared with the control endometrium as follows: miR-16 (*p* < 0.001), miR-20a (*p* < 0.05), miR-21 (*p* < 0.01) and miR-222 (*p* < 0.05). The mRNA levels of VEGF-A and TSP-1 were also increased in the DE lesions compared with the control tissue, although the observed difference was not statistically significant. The menstrual phase did not significantly modify the expression of the angiogenic factors in the DE lesions.

When comparing ectopic with eutopic endometrium from women with DE, the authors identified a significantly increased VEGF-A mRNA from the patients in the proliferative phase of the menstrual cycle (9.66 ± 1.48 vs. 8.07 ± 1.13 (*p* = 0.05)). TSP-1 was also increased in both the proliferative (170 ± 44 vs. 140 ± 28 ng/mg, *p* = 0.05) and secretory (488 ± 303 vs. 69 ± 16 ng/mg, *p* = 0.05) phases of the menstrual cycle [49].

Perricos et al. investigated the peritoneal fluid (PF) of women with endometriosis to identify the biomarker signatures unique to DE [50]. Five biomarkers were significantly overexpressed in the PF of DE patients when compared to the PF of endometriosis patients without DE. These were stem cell factor (1.33-fold, *p* = 0.033), interleukin-6 receptor alpha (1.4-fold, *p* = 0.004), chemokine ligand 19 (1.9-fold, *p* = 0.038), melanoma inhibitory activity (1.32-fold, *p* = 0.040) and VEGF-D (1.75-fold, *p* = 0.034). VEGF-D is a potent growth factor critical in lymphangiogenesis pathways, and thus may be implicated in the lymphatic spread of endometriotic cells in the establishment of the DE lesions.

The lymphagiogenic growth factors VEGF-C and VEGF-D were further evaluated in a study of 38 women undergoing laparoscopic surgery for DE [52]. VEGF-C and VEGF-D expressed moderate to intense staining in epithelial cells and weak to moderate staining in the stromal cells of the DE lesions. The VEGF-C expression was significantly increased in patients without hormonal therapy compared to those with hormonal therapy (*p* < 0.001).

Machado et al. analysed the density and immunolocalisation of VEGF in proliferative and secretory DE lesions compared with eutopic endometrium [43]. The authors did not state which isoform of VEGF was investigated in this study. A total of 62 patients were prospectively recruited for this study, including 30 women with endometriosis and 32 disease-free control patients. The vascular density, the distribution of VEGF and its corresponding receptor VEGFR-2 were found to be significantly higher in the DE lesions. The vessel density in the DE lesions was also significantly higher than the unaffected rectum (10.4 ± 0.89/mm^2^, *p* < 0.05), ovary (4.8 ± 0.75/mm^2^, *p* < 0.05) and bladder (6.3 ± 0.67/mm^2^, *p* < 0.05). The authors also examined the VEGF expression in different endometriotic lesion types and reported DE lesions to have the highest levels (14.5 ± 0.53%) compared with ovarian (13.3 ± 0.48%) and bladder (12.9 ± 0.57%) endometriotic lesions.

In a small study involving 32 patients, Fillipi et al. investigated pro-angiogenic markers in endometriosis using RNA extraction and quantitative real-time PCR [47]. No significant difference in the VEGF-A mRNA levels was observed in the DE lesions compared to control endometrium obtained from women undergoing tubal ligation. Contrasting to Machado et al., the authors report a significantly higher expression of VEGF-A (6-fold, *p* < 0.001) in ovarian endometrioma compared with the DE lesions.

Using Enzyme-Linked Immunosorbent Assays, Bourlev et al. studied the concentrations of VEGF-A, soluble VEGFR-1, VEGFR-2, angiogenin and angiopoietin-2 (Ang-2) in the serum and PF from the healthy controls and DE [51], demonstrating increased levels of VEGF-A in the PF (672 ± 274 pg/mL, *p* < 0.001) and serum (734 ± 156 pg/mL, *p* < 0.05) of the DE patients compared to the controls. The same correlation was found for ANG in the PF (264 ± 76 ng/mL, *p* < 0.05) and the serum (357 ± 84 ng/mL, *p* < 0.05) of the DE patients and for Ang-2 in the serum (1424 ± 254 pg/mL, *p* < 0.05) and PF (6357 ± 938, *p* < 0.001). Surgery to remove DE lesions significantly lowered the serum VEGF-A to 394 ± 123 pg/mL (*p* < 0.05) compared to the pre-surgery levels.

Yerlikaya et al. analysed the mRNA expression levels of 12 angiogenic-associated growth factors, including VEGF-A and VEGFR-2 [45]. They reported increased VEGF-A mRNA expression levels in DE versus control endometrium and eutopic endometrium, although this result did not reach statistical significance. VEGFR-2 was significantly increased in DE compared to control endometrium (*p* < 0.001).

Using IHC, Kim et al. examined the expression of VEGF with an array of other markers such as CD44, matrix metalloproteinase-2 (MMP-2) and Ki-67 in endometriosis [44], and reported that VEGF is present in both glandular epithelium (100%) and stromal cells (81.8%) of the DE lesions; however, there was no statistically significant difference compared to other types of endometriosis. Likewise, there was no significant difference in VEGF staining between stage I/II and stage III/IV endometriosis. The only significant difference found was the lower expression of Ki-67 (11.1%, *p* = 0.002) and MMP-2 (18.2%, *p* = 0.026) in the DE stromal cells. The authors did not state which isoform of VEGF was investigated in this study.

Van Langendonckt et al. used the microarray analysis of excised DE tissue to identify specific markers of DE vasculature [48]. IHC was subsequently used to corroborate the identified expression of six markers including matrix Gla protein (MGP), matrix metalloproteinase-3 (MMP-3), tissue inhibitor of MMP-3 (TIMP-3), desmuslin, desmin and PEP-19 (Purkinje cell protein-4). Despite strong expression in vessels, PEP-19, desmuslin, desmin, TIMP-3 and MMP-3 were not specific for endometriotic vessels because they were also expressed in other cell types. MGP was found to be the most pronounced marker of endometriotic vessels, with an 8.6-fold increase in the fibromuscular tissue vessels (*p* < 0.05) and a 5.9-fold increase in the stromal vessels (*p* < 0.05) compared with the vessels collected from the control tissues.

In a small study of just 41 patients, Stratopoulou et al. used IHC in rectovaginal endometriosis, adenomyosis and control tissue [39], demonstrating that they were all positive for VEGF; however, no significant differences were detected in VEGF staining between the different tissue types studied or between the DE and the control tissue.

### 3.3. HIF-1A Expression

Two studies investigated the HIF-1A mRNA expression in DE [45,47], including laparoscopically excised ectopic endometrial samples and matched eutopic endometrial biopsies collected via hysteroscopies and via dilatation and curettage. Filippi et al. showed no significant difference in the HIF-1A expression between the DE lesions and the control patients, while Yerlikaya et al. demonstrated a significantly higher HIF-1A expression (37.8-fold, *p*< 0.05) in ovarian endometriomas compared to the DE lesions. Yerlikaya et al. also demonstrated that HIF-1A expression in the DE lesions was increased compared to eutopic endometrium; however, this result was insignificant. Similarly, there was no significant difference in the HIF-1A expression between DE and ovarian or peritoneal endometriosis.

## 4. Discussion

This systematic review has collated the evidence on vascular density and the mechanisms of vascularisation in the DE lesions. Compared with other forms of endometriosis, the DE lesions appear to be highly vascularised, as evident by the increased vessel density reported in several studies using the MVD measurements and other analytic methods of vessel density. VEGF was the most widely studied angiogenic factor involved in angiogenesis pertinent to DE lesions, with an overall increase in the VEGF and its associated receptors. The published literature did not identify any significant difference in HIF-1A expression between the DE lesions and the control tissue (Figure 2).

There is wide heterogeneity in the methods employed to examine the vessel density and expression levels of the angiogenic factors. Furthermore, there are several inherent difficulties in examining the tissue and other biosamples from women with DE. DE lesions contain varying degrees of fibrosis, blood vessels, immune cells, the endometrial epithelial and stroma-like cells. When endometriotic lesions are excised, a considerable and variable amount of the surrounding tissue will also be included in such samples. These inconsistencies pose a challenge when analysing excised DE lesions using methods such as qPCR [55], Western blotting and microarrays [56], where the excised lesion has been processed as a whole to extract RNA or proteins. The contribution of the different cell types in such samples to the final analysis is not usually considered.

Endometrial tissues are exquisitely sensitive to ovarian hormones, and their gene and protein expression is, thus, dynamic across the menstrual cycle [57,58,59]. When examining endometrial tissue in ectopic and eutopic locations, standardising the samples according to the menstrual cycle phase is essential to understand the tissue-specific differences [60]. Many studies, however, did not mention the cycle phase of the endometrial or endometriotic lesions studied.

The eutopic endometrium is organised into functionally and morphologically very different layers [61,62]. When comparisons are made between the eutopic endometrium and ectopic lesions, reference to what particular region (functionalis or basalis layer) would make the comparisons more physiologically meaningful. However, almost no study made that distinction.

Many women take hormonal medications to alleviate the symptoms associated with endometriosis. The physiological alteration of hormones according to the menstrual cycle will also affect the angiogenic markers and the MVD levels in endometrium-like tissues. Most studies have disregarded these crucial aspects, and there is a lack of standardisation in sample collection and analysis. For these reasons combined, drawing robust conclusions from the data presented in this systematic review regarding the presence or absence of abnormal vascularisation of DE is challenging. The recent efforts by organisations such as ESHRE and WEC have proposed standardising the sample collection with valuable clinical and symptom-related data, which will reduce this limitation in future studies [63,64].

### 4.1. Microvessel Density

MVD is a common measure used to evaluate the vasculature of tissues, reflecting the amount of angiogenesis [65]. The majority of studies assessing the MVD found that the DE lesions had a greater MVD, indicating that they are more vascularised than superficial forms of endometriosis and control endometrium. DE lesions infiltrate beyond the peritoneal lining of intra-abdominal organs. Therefore, these findings corroborate the current understanding that endometriotic lesions’ invasiveness is positively correlated with the degree of vascularisation.

The authors studied two major markers of endothelial cell vascularisation, CD31 and CD34. CD31, also known as Platelet endothelial cell adhesion molecule (PECAM-1), is a transmembrane glycoprotein marker expressed in endothelial cells and haematopoietic cells, and thus, may indicate the mere presence of endothelial cells. CD34, on the other hand, is a transmembrane glycoprotein expressed on endothelial cells and plays a role in endothelial cell interaction and adhesion, and therefore has an essential role in angiogenesis [38]; its presence indicates the angiogenic process occurring in the tissues.

The anti-angiogenic effects of progestins in DE have been highlighted with progestin treatment demonstrating to significantly decrease the MVD of the DE lesions. This anti-angiogenic effect of progestin is similar to what is seen in endometrial cancer, where progestins can inhibit angiogenesis and, in turn, inhibit tumour progression [66]. Additionally, it supports the continued use of progestin therapy in the treatment of endometriosis, including DE, despite the anti-angiogenic benefit being more significant in other forms of the disease [67].

### 4.2. VEGF

VEGF expression and, in particular, its isoform VEGF-A, is significantly higher in DE than in other endometriotic phenotypes and control endometrium. Correspondingly, there is also an increased expression of the VEGFR-2 receptor.

Angiogenesis has been evidenced to be the greatest in rectal endometriosis compared to bladder endometriosis [43]. Both bladder and rectal diseases are considered “DE”, so the difference in the degree of angiogenesis signifies the heterogeneity of endometriotic lesions. Different patterns of vascularisation observed in different organs potentially signify the diverse mechanisms of infiltration and growth amongst endometriosis subtypes.

Several authors have described angiogenesis as an essential step in the implantation and development processes through which endometrial tissue enters the peritoneal cavity, with VEGF as one of the most critical mediators [22,47,65]. Machado et al. found both VEGF and its receptor, VEGFR-2, to be increased in DE compared to other types of endometrioses. The binding of VEGF to VEGFR-2 is associated with neovascularisation [43]. The high serum and peritoneal levels of VEGF-A, ANG and Ang-2 have been reported in DE patients [51]. All three factors are known to facilitate angiogenesis. They found that the surgical removal of visible endometriosis reduced the serum VEGF-A levels postoperatively [51], thus suggesting that the endometriotic lesions were the source of the increased levels. It is unknown if the removal of the DE lesions improved patient symptoms. To ascertain this information, an extended postoperative follow-up period should be included in future studies.

Interestingly, only one paper found VEGF-A expression to be similar between DE and control endometrium [47]. The authors found a significant increase of VEGF-A mRNA in ovarian endometriosis but not DE, thus in contrast to all other studies included in this review. The authors did not provide an explanation for their contrasting findings but referred to a paper which examined superficial peritoneal endometriosis, implying that the second paper supports their findings [47,68]. Therefore, there is no apparent explanation for the reported difference by Filippi et al. to confirm the credibility of their data.

The variation in VEGF expression levels amongst the different phenotypes of endometriosis supports the hypothesis that such conditions may contribute to the development of endometriosis but not to its maintenance. Similarly, Kim et al. found no significant difference in the VEGF expression between the studied phenotypes of endometriosis [44]. In DE, the stromal cells are not actively proliferating as they are in, for example, peritoneal endometriosis. The inactivity of stromal cells is perhaps due to the fibrotic nature of DE, rendering the endometriotic lesions “burnt out” and inactive [69]. Stromal cells are essential for the invasion into the adjacent host tissue [44]. Additionally, the stromal cells expressed lower VEGF in rectovaginal DE compared to peritoneal and ovarian endometriosis. The lower expression may be explained by the fact that stromal cells produce VEGF and other factors that mediate the activity of vascular endothelial cells [70,71].

Consequently, vascular activity can be expected to be reduced due to the lack of need for further invasion and growth. If so, the vascularisation of active and inactive endometriosis is incomparable. However, none of the included studies clearly examined the amount of fibrosis, inflammation and presence of different cell types (e.g., stromal/epithelial) in the DE samples to confirm the differences observed in the results.

Establishing an adequate blood supply for the growth and maintenance of endometriotic lesions is essential during initial implantation [47]. Angiogenesis, therefore, is most critical during the early stages of development. However, the studies in this review are limited by the lack of “early” lesions and small DE sample sizes [47]. At the point of analysis, DE lesions are well established beyond the uterus, causing symptomatology, and are usually associated with a delayed presentation to health services. The variation in the age of lesions makes it challenging to ascertain the processes actively occurring in the samples at the time of collection [39]. Although the growth of DE may be slow, the disease is still progressing with continuing inflammation and fibrosis, nonetheless.

VEGF-C and VEGF-D, critical growth factors in lymphangiogenesis, are increased in DE. Lymphangiogenesis is a process central to malignant tumour development, infiltration and metastasis. Malignant tumours have a high VEGF-C expression and an increased lymph vessel density, facilitating the metastatic spread of cancerous cells [72,73,74,75]. DE shares phenotypical similarities with malignant tumours regarding its ability to infiltrate surrounding structures. We postulate that dysfunctional lymphangiogenesis may contribute to DE’s ability to infiltrate adjacent organs.

It is interesting to note the negative correlation between the expression of VEGF-C and hormonal therapy, which reflects the negative correlation between VEGF-A and progestin therapy. Hormonal therapies are the cornerstone of the management of all forms of endometriosis. The manipulation of VEGF-A and VEGF-C through hormonal therapies adds to the explanation and mechanism of hormonal therapies for treating DE.

### 4.3. HIF-1A

Since the pathological function of HIF-1A in endometriosis was discovered in 2007, the relationship between the pathogenesis of the disease and hypoxic stress has become increasingly evident [76]. Recently, Wu et al. highlighted steroidogenesis, angiogenesis and epigenetic regulation as the three main hypoxia-influenced processes in endometriosis [77]. In all three processes, HIF-1A expression was upregulated. HIF-1A is, therefore, a recognised angiogenic factor in the pathogenesis of endometriosis that causes the necessary neovascularisation of endometriotic lesions in hypoxic environments [33,78]. The cyclical change in HIF-1A in the endometrium is well established [79], and both included studies considered the menstrual cycle phase of recruited patients.

Despite the existing evidence, this review’s findings suggest that HIF-1A mRNA does not play an additional role in the vascularisation of DE lesions [45,47]. Goteri et al. reported HIF-1A expression to be higher in endometriomas [80]. It is noteworthy that the HIF-1A protein, but not the mRNA, is likely to correlate with hypoxic activity. Therefore, future studies should examine the protein levels to conclude the involvement of HIF-1A in DE.

Becker et al. offered a possible explanation for the insignificant expression of HIF-1A in DE. In this study utilising Western blotting, HIF-1A protein was only increased during the early stages of transplantation and invasion of endometriotic lesions, in turn, suggesting that HIF-1A may have a role in the initiation of superficial disease but not in the progression to DE or maintenance of DE [81]. It is possible that in the highly oxygenated environment of established endometriotic lesions, with a mature and dense microvascular network, HIF-1A is no longer expressed.

### 4.4. Strengths and Limitations

The findings from this study are strengthened by the broad inclusion parameters. The review involved a comprehensive search string and included human and animal studies of all languages to ensure that all the potentially relevant papers were screened. Moreover, the multiple databases that were searched and the chaining processes reduced the likelihood of missing relevant papers.

The main limitation of this study was the heterogeneity of the methods and experimental techniques used in the included papers. Researchers disregarding the menstrual cycle phase, the different endometrial sub-regions they study, and the different proportion of cell types that a lesion, which they are analysing, may contain, can affect the results and translatability of the data. This reduced the comparability of the results and, in turn, restricted the conclusions that could be drawn. The lack of consistency in the experimental methods is likely due to the small pool of research available on the topic, meaning there is no established precedent. This can be improved with further research on the vascularisation of DE.

## 5. Conclusions

Understanding the pathophysiology of DE remains a challenge. This review has consolidated the understanding that DE lesions are highly vascularised, differing from normal endometrium and other types of endometrioses. Furthermore, VEGF has been highlighted as a critical factor in the process of angiogenesis, with high levels of VEGF and its receptor expressed in DE. Angiogenesis must occur at a significant rate to result in the high MVDs reported in this review. Therefore, VEGF may be implicated as an appealing therapeutic target. For this reason, more studies with larger sample sizes of DE are required with a standardised methodology to support these conclusions to develop new, non-invasive therapeutic options outside of surgery.

## Figures and Tables

**Figure 1 cells-12-01318-f001:**
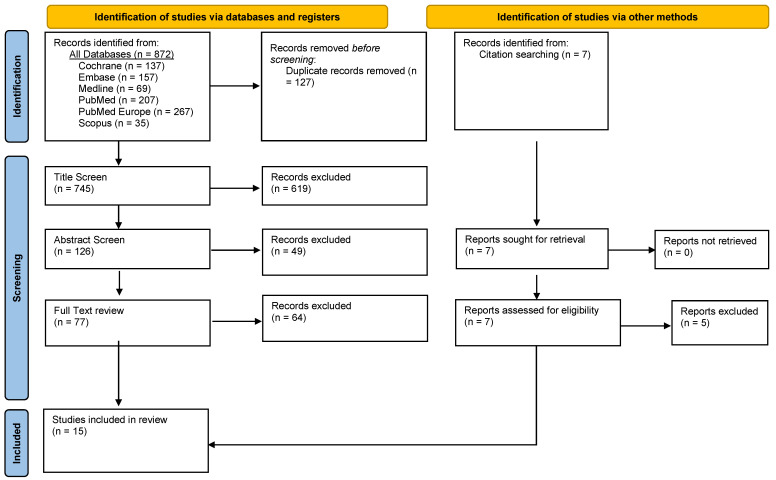
PRISMA flowchart demonstrating the selection of publications identified in the systematic review of the literature.

**Figure 2 cells-12-01318-f002:**
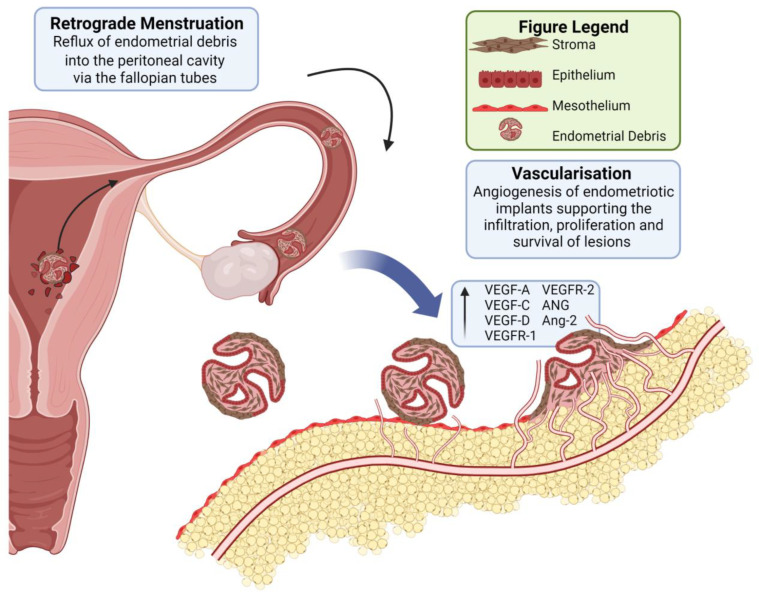
Vascularisation changes associated with the implantation of deep endometriotic lesions.

**Table 1 cells-12-01318-t001:** Study characteristics of studies conducted using IHC experimental techniques.

Author (Year)	Focus of Study	Experimental Technique	Number of Patients				Hormonal Treatment	Phase of Menstrual Cycle	Relevant Findings of Study
			Superficial Endometriosis	DE	Control (Description of Control Cohort)	Total			
Robin 2016 [41]	MVD	Immunohistochemistry FSHR marker	8	186	17 (healthy uterine tissue obtained from patients undergoing benign gynaecological procedures. Histological examination confirmed the absence of endometrial pathology)	211	Not Stated	Matched proliferative and secretory phases. Statistical analysis of samples was undertaken according to menstrual phase	Increased FSHR-positive MVD in DE and superficial endometriosis lesions compared to control samples. Similar MVD in DE and superficial endometriosis *
Stratopoulou 2021 [39]	MVD VEGF	Immunohistochemistry VEGF, CD31 and αSMA staining of microvessels	0	13	14 (healthy uterine tissue obtained from patients undergoing benign gynaecological procedures. Histological examination confirmed the absence of endometrial pathology)	41	None had hormone therapy for at least three months prior to surgery	Samples were collected throughout proliferative, secretory and menstrual phases in an unmatched manner. No statistical analysis of samples from different menstrual phases was undertaken	Increased CD31-positive MVD in DE compared to control tissue. No difference in VEGF staining intensity between DE, superficial endometriosis and control tissue *
Machado 2008 [43]	MVD VEGF	Immunohistochemistry assessment of blood vessels using VEGF, vWF and Flk-1	10	20	20 (healthy uterine tissue obtained from patients undergoing benign gynaecological procedures. Histological examination confirmed the absence of endometrial pathology) 12 (normal tissues of ovary (n = 4), bladder (n = 4) and rectum (n = 4) were obtained from these organs beside the endometriotic lesions)	62	None of the patients received hormonal treatment for at least three months before the study	Matched proliferative and secretory phases. Statistical analysis of samples was undertaken according to menstrual phase. 51	Increased wVF-positive MVD in DE compared to control endometrium (in both proliferative and secretory phases) and control rectal tissue
Keichel 2011 [52]	VEGF	Immunohistochemistry VEGF-C & VEGF-D	0	38	13 (tumour-free marginal border of the resection part of patients with colon cancer) 10 (unaffected vagina taken during hysterectomy due to benign diseases)	61	Patients receiving and not receiving hormonal contraceptives were recruited for this study	Samples were collected throughout proliferative, secretory and menstrual phases in an unmatched manner. No statistical analysis of samples from different menstrual phases was undertaken	Increased staining of VEGF-C and VEGF-D and lymph vessel density in DE
Signorile 2009 [40]	MVD	Immunohistochemistry CD34, PR, ER markers	6	56	0	62	Not Stated	Not Stated	Intense CD34 staining associated with DE lesions compared to superficial endometriosis. ER and PR expression in DE lesions correlated to the degree of vascularisation
Jondet 2006 [46]	MVD	Immunohistochemistry assessment of blood vessels using CD31	32	34	0	66	Study included both progestin- and non-progestin-treated women	Not Stated	Increased CD31 staining associated with DE lesions compared to superficial endometriosis Progestin therapy significantly reduces MVD in both eutopic endometrium and endometriotic lesions
Vinci 2016 [38]	MVD	Immunohistochemistry CD31 staining of vessels within endometriotic “hot spots”	0	113	0	113	All were treated with GnRH agonists at least six months before surgery	Not Stated	CD31 staining was found to be “intense” in 42% of the DE tissue samples, “medium” in 32% and “weak” in 26% of DE “hot spots.”
Raimondo 2020 [42]	MVD	Intraoperative Indocyanine green angiography assessment of endometriotic lesions. Immunohistochemistry assessment of endometriotic biopsies using CD31	0	30	0	30	All women *assumed* progestin therapy in the three months before surgery	Not Stated	IGA assessment identified 60% of the DE lesions were found to be hypovascular, while the remaining 40% were deemed to be hypervascular Using CD31 staining, hypovascular lesions were demonstrated to have a significantly lower MVD than hypervascular lesions
Kim 2007 [44]	VEGF	Immunohistochemistry of stroma using CD44, VEGF and Ki-67	51	11	0	62	None of the patients received hormonal treatment for at least six months before the study	Not Stated	No difference in VEGF staining between stages I/II and III/IV of endometriosis lesions *

* Result not of statistical significance.

**Table 2 cells-12-01318-t002:** Study characteristics of studies conducted using qPCR and other experimental techniques.

Author (Year)	Focus of Study	Experimental Technique	Number of Patients				Hormonal Treatment	Phase of Menstrual Cycle	Relevant Findings of Study
			Superficial Endometriosis	DE	Control (Description of Control Cohort)	Total			
Van Langendonckt 2007 [48]	MVD	Laser Capture Microdissection and qPCR	0	28	20 (eutopic endometrium obtained from recruited patients with DE. No histological evidence of endometrial pathology was identified in the control tissue as assessed by a pathologist)	28	Patients were not receiving hormonal treatment at the time of tissue collection	Samples were collected throughout proliferative, secretory and menstrual phases in an unmatched manner. No statistical analysis of samples from different menstrual phases was undertaken	Significantly raised expression of MGP in DE microvessels compared with vessels collected from control tissues
Perricos 2020 [50]	VEGF	Untargeted analysis of 92 cancer-related proteins (including VEGF-D) using the Proseek Multiplex Oncology I Cancer Panel	34	19	31 (Peritoneal fluid obtained from patients undergoing benign gynaecological procedures without macroscopic evidence of endometriosis at the time of surgery)	84	Not Stated	Samples were collected throughout proliferative and secretory phases in an unmatched manner. No statistical analysis of samples from different menstrual phases was undertaken	Increased VEGF-D in PF of DE patients compared to superficial endometriosis
Bourlev 2010 [51]	VEGF	Peritoneal and serum fluid analysis of VEGF-A, VEGFR-1 and Ang-2 using ELISA	0	32	21 (serum and peritoneal fluid collected from healthy patients undergoing laparoscopic sterilisation)	53	None of the patients received hormonal treatment for at least three months before the study	Samples were collected throughout proliferative, secretory and menstrual phases in an unmatched manner. No statistical analysis of samples from different menstrual phases was undertaken	Increased concentration of VEGF-A in serum and peritoneal fluid of patients with DE compared to controls. Surgery to remove DE lesions significantly lowered serum VEGF-A
Ramón 2011 2011 [49]	VEGF	VEGF-A mRNA expression was assessed using qPCR. VEGF-A protein levels were quantified by ELISA	45	13	38 (healthy uterine tissue obtained from patients undergoing benign gynaecological procedures. Histological examination confirmed the absence of endometrial pathology)	96	None of the patients received hormonal treatment for at least three months before the study	Samples were collected throughout proliferative and secretory phases. Statistical analysis of samples was undertaken according to menstrual phase	Increased levels of VEGF-A of DE lesions compared with control tissue *
Filippi 2016 [47]	VEGF HIF-1A	VEGF-A and HIF-1A expression was assessed using qPCR	16	11	15 (healthy uterine tissue obtained from patients undergoing hysteroscopy)	42	None of the patients received hormonal treatment for at least three months before the study	Proliferative samples only	No significant difference in the VEGF-A mRNA levels in DE lesions compared to control endometrium *
Yerlikaya 2016 [45]	VEGF HIF-1A	VEGFA, VEGFR2, HIF-1A and ANG mRNA expression was analysed using qPCR	23	38	53 (healthy uterine tissue obtained from patients undergoing benign gynaecological procedures. Histological examination confirmed the absence of endometrial pathology)	114	Not Stated	Samples were collected throughout proliferative and secretory phases. Statistical analysis of samples was undertaken according to menstrual phase	Increased VEGF-A mRNA expression levels in DE versus control endometrium and eutopic endometrium *

* Result not of statistical significance.

**Table 3 cells-12-01318-t003:** Risk of bias assessment—Newcastle–Ottawa Scale (NOS).

Paper Reference	Adequate Case Definition?	Representativeness of the Cases?	Selection of Controls?	Definition of Controls?	Comparability of Cases and Controls on the Basis of the Design or Analysis?	Ascertainment of Exposure?	Same Method of Ascertainment for Cases and Controls?	Non-Response Rate?	Selection	Comparability	Exposure	Total Score	Overall Quality
Vinci 2016 [38]	*	0	*	*	0	0	0	0	3	0	0	3	Poor
Van Langendonckt 2007 [48]	*	*	*	*	*	*	*	*	4	1	3	8	Good
Raimondo 2020 [42]	*	*	*	0	*	*	*	*	3	1	3	7	Good
Signorile 2009 [40]	*	*	0	0	0	0	0	0	2	0	0	2	Poor
Robin 2016 [41]	*	*		*	*	*	0	*	3	1	2	6	Good
Jondet 2006 [46]	*	0	*	*	0	*	*	*	3	0	3	6	Poor
Stramiddleoulou 2021 [39]	*	*	*	*	*	*	*	*	4	1	3	8	Good
Machado 2008 [43]	*	0	*	*	*	*	*	*	3	1	3	7	Good
Keichel 2011 [52]	*	*	*	*	**	*	*	*	4	2	3	9	Good
Ramón 2011 [49]	*	*	*	*	*	*	0	*	4	1	2	7	Good
Perricos 2020 [50]	*	*	*	*	*	*	0	*	4	1	2	7	Good
Bourlev 2010 [51]	*	0	*	*	**	*	*	*	3	2	3	8	Good
Filippi 2016 [47]	*	*	*	*	*	*	*	*	4	1	3	8	Good
Yerlikaya 2016 [45]	*	0	*	0	**	*	*	*	2	3	3	7	Good
Kim 2007 [44]	*	0	*	0	*	*	*	*	2	1	3	6	Fair

* = 1 point scored, 0 = no points scored. Thresholds for converting the Newcastle–Ottawa Scale to Agency for Health Research and Quality (AHRQ) standards (good, fair and poor) as follows: Good quality (Green): 3 or 4 stars in selection domain AND 1 or 2 stars in comparability domain AND 2 or 3 stars in outcome/exposure domain. Fair quality (Yellow): 2 stars in selection domain AND 1 or 2 stars in comparability domain AND 2 or 3 stars in outcome/exposure domain. Poor quality (Red): 0 or 1 star in selection domain OR 0 stars in comparability domain OR 0 or 1 star in outcome/exposure domain.

## Data Availability

Data sharing not applicable. No new data were created or analysed in this study. Data sharing is not applicable to this article.

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
