# Peer review of "Vascularisation in Deep Endometriosis: A Systematic Review with Narrative Outcomes"

_cells, 2023, doi:10.3390/cells12091318_

Round 1

Reviewer 1 Report

In this study, the authors reviewed the available literature, chosen with very narrow and clearly described criteria and broad inclusion parameters, focusing on the Deep Endometriosis-specific vascularization, including microvessel density, VEGF and HIF-1A expression.

The authors also reported that, if the abnormal vascularization has been reported to be related to the pathogenesis of DE, for the remaining factors the wide heterogeneity of methods and experimental techniques may be the cause of the lack of consistency of the results analyzed, thus making the conclusions even more difficult.

The topic of this review is very interesting and has been tackled rigorously and effectively. Precisely because of clinical interest, I would suggest to the authors to make a further effort of analysis and, above all, of synthesis, trying to develop a flow diagram capable of summarizing the data obtained on the 3 parameters (MVD, VEGF and HIF-1A), possibly underlining their variability.

Correct and clearly structured and understandable language

Author Response

Thank you for reviewing our manuscript and providing your valuable feedback. As requested, we have produced two new figures to summarise and highlight pathological alterations in vascularisation associated with deep endometriosis.

Reviewer 2 Report

1.      This is a well-written and well-researched paper. The authors present a systemic review of the evidence for deep endometriosis (DE) Pacific vascular is Asian. The lesions were highly vascularized. VEGF-a, VEGF-B, and VEGFR to were significantly raised. Progestin therapy was associated with a decrease in microvascular density. Abnormal vascularization is a potential therapeutic target.

2.      Line 34: Linda Griffith would disagree with your use of benign and say "Please don't refer to endometriosis, adenomyosis, or fibroids as 'benign disease'-nope, not benign, they are 'common and morbid'". Mark Noar calls them “the cancer that does not kill.” The phrase "nonmalignant but morbid" is okay but still understates the problem for some women. "Benign," although syntactically correct, is misleading.

3.      Line 38: Laganà et al. will be disappointed that you do not like their "unus pro omnibus, omnes pro uno” unifying theory. [PMID 28571791]" Consider changing "and a unifying theory..." to "and a universally accepted unifying theory..." I agree with you, I do not completely accept Lagana either, but he has a brilliant productive mind and may be correct.

Laganà AS, Vitale SG, Salmeri FM, Triolo O, Ban Frangež H, Vrtačnik-Bokal E, Stojanovska L, Apostolopoulos V, Granese R, Sofo V. Unus pro omnibus, omnes pro uno: A novel, evidence-based, unifying theory for the pathogenesis of endometriosis. Med Hypotheses. 2017 Jun;103:10-20. doi: 10.1016/j.mehy.2017.03.032. Epub 2017 Mar 31. PMID: 28571791.

4.      Line 39: Although Sampson is the most accepted, it is not the longest-standing. Cruveilhier (1835), Waldeyer (1870), Marchand (1879), Cullen (1896), Russell (1899), and many more are candidates for the longest-standing concept.

5.      Line 39: The Samson article you quote discusses misplaced endometrial-like tissue, Müllerian tissue, pseudo-endometrial tissue, metaplasia of peritoneum, and possibly other sources but does not appear to discuss retrograde dissemination. Simpson's first mention of retrograde was in the amended note on pages 235 and 236 of his near duplicate article in the Transactions of the American Gynecologic Society. He expanded that in his Sentinel paper in 1927.

Adapted from Batt "A History of Endometriosis" 2011: The first occasion that Sampson alluded to his transplantation theory is published in the Transactions of the American Gynecological Society (Trans Am Gynecol Soc 1921;46:162 236) version of "Perforating hemorrhagic (chocolate) cysts of the ovary. Their importance and especially their relation to pelvic adenomas of the endometrial type ("adenomyoma" of the uterus, rectovaginal septum, sigmoid, etc.)." That article was published in Arch Surg (now JAMA Surgery). 1921, 3(2):245-323 without the amended note that is on pages 235-236. The Transactions of the American Gynecological Society version, page 236, has:

"Two possible sources of the origin of these small tubules or cysts of endometrial type in the ovary present themselves: first, congenital, and second, acquired from the implantation of epithelium escaping from the tube during menstruation and its subsequent invasion of the ovary."

https://www.google.com/books/edition/Transactions_of_the_American_Gynecologic/

QqFEAAAAYAAJ?hl=en&gbpv=1&bsq=%22Two%20possible%20sources%20of%20the%

20origin%22

Batt RE. A History of Endometriosis. Springer-Verlag London Ltd., London, 2011.

https://www.springer.com/us/book/9780857295842

https://www.google.com/books/edition/A_History_of_Endometriosis/JyoywyVfIhkC?hl=en&gbpv=0

https://epdf.pub/queue/a-history-of-endometriosis.html

Sampson JA. Perforating hemorrhagic (chocolate) cysts of the ovary: their importance and especially their relation to pelvic adenomas of the endometrial type ("adenomyoma" of the uterus, rectovaginal septum, sigmoid, etc.) Trans Am Gynecol Soc 1921;46:162-241.

The 1921 Transactions of the American Gynecological Society issues are at Google Books:

https://www.google.com/books/edition/Transactions_of_the_American_Gynecologic/QqFEAAAAYAAJ?hl=en&gbpv=1

Sampson JA. Peritoneal endometriosis due to the menstrual dissemination of endometrial tissue into the peritoneal cavity. Am J Obstet Gynecol. 1927, 14(4):422-469, doi: 10.1016/S0002-9378(15)30003-X.

https://doi.org/10.1016/S0002-9378(15)30003-X

6.      Line 49: consider adding immune dysfunction, escape from immunologic control, and epigenetic factors to mechanisms.

7.      Line 50-368: The main body of the paper is enjoyable to read, comprehensive, informative, and should be published promptly.

Author Response

  1. This is a well-written and well-researched paper. The authors present a systemic review of the evidence for deep endometriosis (DE) Pacific vascular is Asian. The lesions were highly vascularised. VEGF-a, VEGF-B, and VEGFR to were significantly raised. Progestin therapy was associated with a decrease in microvascular density. Abnormal vascularisation is a potential therapeutic target.

Many thanks for appreciating our work and for the very informative and constructive review of the work. We are particularly grateful for updating our knowledge on some interesting publications and the history of endometriosis.

  1. Line 34: Linda Griffith would disagree with your use of benign and say "Please don't refer to endometriosis, adenomyosis, or fibroids as 'benign disease'-nope, not benign, they are 'common and morbid'". Mark Noar calls them "the cancer that does not kill." The phrase "nonmalignant but morbid" is okay but still understates the problem for some women. "Benign," although syntactically correct, is misleading.

We agree and have revised the manuscript accordingly.

Endometriosis-associated symptoms cause a huge burden on women and society, and non-of the available therapies are universally effective or curative. There are fundamental issues with our understanding of the disease and the causative relationship of the disease with the associated symptoms, which hamper the development of new, more effective treatments. Therefore, revising the existing data and deciphering the important avenues to progress for therapeutic targets is an important aspect of endometriosis research.

  1. Line 38: Laganà et al. will be disappointed that you do not like their "unus pro omnibus, omnes pro uno" unifying theory. [PMID 28571791]" Consider changing "and a unifying theory..." to "and a universally accepted unifying theory..." I agree with you, I do not completely accept Lagana either, but he has a brilliant productive mind and may be correct.

Laganà AS, Vitale SG, Salmeri FM, Triolo O, Ban Frangež H, Vrtačnik-Bokal E, Stojanovska L, Apostolopoulos V, Granese R, Sofo V. Unus pro omnibus, omnes pro uno: A novel, evidence-based, unifying theory for the pathogenesis of endometriosis. Med Hypotheses. 2017 Jun;103:10-20. doi: 10.1016/j.mehy.2017.03.032. Epub 2017 Mar 31. PMID: 28571791.

We agree, Laganà et al. propose a unifing theory which many are speculative of due to lack of robust evidence. We have revised the manuscript to include this

  1. Line 39: Although Sampson is the most accepted, it is not the longest-standing. Cruveilhier (1835), Waldeyer (1870), Marchand (1879), Cullen (1896), Russell (1899), and many more are candidates for the longest-standing concept.

For clarity, we have rephrased the relevant section to state that Sampson's theory is the most known, not the longest-standing.

  1. Line 39: The Samson article you quote discusses misplaced endometrial-like tissue, Müllerian tissue, pseudo-endometrial tissue, metaplasia of peritoneum, and possibly other sources but does not appear to discuss retrograde dissemination. Simpson's first mention of retrograde was in the amended note on pages 235 and 236 of his near duplicate article in the Transactions of the American Gynecologic Society. He expanded that in his Sentinel paper in 1927.

Adapted from Batt "A History of Endometriosis" 2011: The first occasion that Sampson alluded to his transplantation theory is published in the Transactions of the American Gynecological Society (Trans Am Gynecol Soc 1921;46:162 236) version of "Perforating hemorrhagic (chocolate) cysts of the ovary. Their importance and especially their relation to pelvic adenomas of the endometrial type ("adenomyoma" of the uterus, rectovaginal septum, sigmoid, etc.)." That article was published in Arch Surg (now JAMA Surgery). 1921, 3(2):245-323 without the amended note that is on pages 235-236. The Transactions of the American Gynecological Society version, page 236, has:

"Two possible sources of the origin of these small tubules or cysts of endometrial type in the ovary present themselves: first, congenital, and second, acquired from the implantation of epithelium escaping from the tube during menstruation and its subsequent invasion of the ovary."

https://www.google.com/books/edition/Transactions_of_the_American_Gynecologic/

QqFEAAAAYAAJ?hl=en&gbpv=1&bsq=%22Two%20possible%20sources%20of%20the%

20origin%22

Batt RE. A History of Endometriosis. Springer-Verlag London Ltd., London, 2011.

https://www.springer.com/us/book/9780857295842

https://www.google.com/books/edition/A_History_of_Endometriosis/JyoywyVfIhkC?hl=en&gbpv=0

https://epdf.pub/queue/a-history-of-endometriosis.html

Sampson JA. Perforating hemorrhagic (chocolate) cysts of the ovary: their importance and especially their relation to pelvic adenomas of the endometrial type ("adenomyoma" of the uterus, rectovaginal septum, sigmoid, etc.) Trans Am Gynecol Soc 1921;46:162-241.

The 1921 Transactions of the American Gynecological Society issues are at Google Books:

https://www.google.com/books/edition/Transactions_of_the_American_Gynecologic/QqFEAAAAYAAJ?hl=en&gbpv=1

Sampson JA. Peritoneal endometriosis due to the menstrual dissemination of endometrial tissue into the peritoneal cavity. Am J Obstet Gynecol. 1927, 14(4):422-469, doi: 10.1016/S0002-9378(15)30003-X.

https://doi.org/10.1016/S0002-9378(15)30003-X

Thank you for providing this fascinating perspective on the history of endometriosis. We have amended the references in order to cite Sampson’s 1921 paper “Perforating hemorrhagic (chocolate) cysts of the ovary: their importance and especially their relation to pelvic adenomas of the endometrial type ("adenomyoma" of the uterus, rectovaginal septum, sigmoid, etc.)”

  1. Line 49: consider adding immune dysfunction, escape from immunologic control, and epigenetic factors to mechanisms.

Immune dysfunction and epigenetic factors are pivotal to the pathogenesis of endometriosis. We have amended the manuscript to recognise these factors' crucial role.

  1. Line 50-368: The main body of the paper is enjoyable to read, comprehensive, informative, and should be published promptly.

We thank the reviewer for their kind comments